Full depth CNN classifier for handwritten and license plate characters recognition

http://orcid.org/0000-0002-2913-7671 Salemdeeb Mohammed 1 msalemdeeb@bartin.edu.tr
Ertürk Sarp 2
1 Department of Electrical-Electronics Engineering, Bartin University , Bartin , Turkey
2 Department of Electronics & Communication Eng., Kocaeli University , Izmit, Kocaeli , Turkey
Santosh Kc
Electronic publication date: 2021 Jun 18
Publication date: 2021
Volume: 7
Electronic Location ID: e576
Received 2020 Oct 27; Accepted 2021 May 12
Copyright: © 2021 Salemdeeb and Ertürk
Copyright year: 2021
Copyright holder: Salemdeeb and Ertürk
License: This is an open access article distributed under the terms of the Creative Commons Attribution License, which permits unrestricted use, distribution, reproduction and adaptation in any medium and for any purpose provided that it is properly attributed. For attribution, the original author(s), title, publication source (PeerJ Computer Science) and either DOI or URL of the article must be cited.
License URL: https://creativecommons.org/licenses/by/4.0/

Keywords: Convolutional neural nework, Character recognition, License plate character recognition, Arabic license plate character recognition, Arabic character recognition, Handwritten character recognition, Deep learning, Image classififcation

Funding: The authors received no funding for this work.

==============================
Character recognition is an important research field of interest for many applications. In recent years, deep learning has made breakthroughs in image classification, especially for character recognition. However, convolutional neural networks (CNN) still deliver state-of-the-art results in this area. Motivated by the success of CNNs, this paper proposes a simple novel full depth stacked CNN architecture for Latin and Arabic handwritten alphanumeric characters that is also utilized for license plate (LP) characters recognition. The proposed architecture is constructed by four convolutional layers, two max-pooling layers, and one fully connected layer. This architecture is low-complex, fast, reliable and achieves very promising classification accuracy that may move the field forward in terms of low complexity, high accuracy and full feature extraction. The proposed approach is tested on four benchmarks for handwritten character datasets, Fashion-MNIST dataset, public LP character datasets and a newly introduced real LP isolated character dataset. The proposed approach tests report an error of only 0.28% for MNIST, 0.34% for MAHDB, 1.45% for AHCD, 3.81% for AIA9K, 5.00% for Fashion-MNIST, 0.26% for Saudi license plate character and 0.97% for Latin license plate characters datasets. The license plate characters include license plates from Turkey (TR), Europe (EU), USA, United Arab Emirates (UAE) and Kingdom of Saudi Arabia (KSA).

Introduction

Character recognition (CR) plays a key role in many applications and motivates R&D in the field for accurate and fast classification solutions. CR has been widely investigated in many languages using different proposed methods. In the last years, researchers widely used CNN as deep learning classifiers and achieved good results on handwritten Alphanumeric in many languages (Lecun et al., 1998; Abdleazeem & El-Sherif, 2008; El-Sawy, Loey & EL-Bakry, 2017), character recognition in real-world images (Netzer et al., 2011), document scanning, optical character recognition (OCR) and automatic license plate character recognition (ALPR) (Comelli et al., 1995). Searching for text information in images is a time-consuming process that largely benefits of CR. The connectivity of letters makes classification a challenge, particular for the Arabic language (Eltay, Zidouri & Ahmad, 2020). Therefore, isolated character datasets get more interest in research.

MNIST is a handwritten digits dataset introduced by Lecun et al. (1998) and used to test supervised machine learning algorithms. The best accuracy obtained by stacked CNN architectures, until before two years, is a test error rate of 0.35% in Cireşan et al. (2010), where large deep CNN of nine layers with an elastic distortion applied to the input images. Narrowing the gap to human performance, a new architecture of five committees of seven deep CNNs with six width normalization and elastic distortion was trained and tested in Ciresan et al. (2011) and reported an error rate of 0.27%, where the main CNN is seven stacked layers. In Ciregan, Meier & Schmidhuber (2012), a near-human performance error rate of 0.23% was achieved, where several techniques were combined in a novel way to build a multi-column deep neural network (MCDNN) inspired by micro-columns of neurons in cerebral cortex compared to the number of layers found between retina and visual cortex of macaque monkeys.

Recently, Moradi, Berangi & Minaei (2019) developed a new CNN architecture with orthogonal feature maps based on Residual modules of ResNet (He et al., 2016) and Inception modules of GoogleNet (Szegedy et al., 2015), with 534,474 learnable parameters which are equal to SqeezeNet (Iandola et al., 2016) learnable parameters, and thus the model reported an error of 0.28%. However, a CNN architecture for small size input images of 20 × 20 pixels was proposed in Le & Nguyen (2019). In addition, a multimodal deep learning architecture was proposed in Kowsari et al. (2018), where deep neural networks (DNN), CNN and recurrent neural networks (RNN) were used in one architecture design achieving an error of 0.18%. A plain CNN with stochastic optimization method was proposed in Assiri (2019), this method applied regular Dropout layers after each pooling and fully connected (FC) layers, this 15 stacked layers approach obtained an error of 0.17% by 13.21M parameters. Hirata & Takahashi (2020) proposed an architecture with one base CNN and multiple FC sub-networks, this 28 spars layers architecture with 28.67M parameters obtained an error of 0.16%. Byerly, Kalganova & Dear (2020) presented a CNN design with additional branches after certain convolutions, and from each branch, they transformed each of the final filters into a pair of homogeneous vector capsules, this 21 spars layers obtained an error of 0.16%.

While MNIST was well studied in the literature, there were only a few works on Arabic handwritten character recognition (Abdleazeem & El-Sherif, 2008). The large Arabic Handwritten Digits (AHDBase) has been introduced in El-Sherif & Abdelazeem (2007). Abdleazeem & El-Sherif (2008) modified AHDBase to be MADBase and evaluated 54 different classifier/features combinations and reported a classification error of 0.52% utilizing radial basis function (RBF) and support vector machine (SVM). Also, they discussed the problem of Arabic zero, which is just a dot and smaller than other digits. They solved the problem by introducing a size-sensitive feature which is the ratio of the digit bounding box area to the average bounding box area of all digits in AHDBase’s training set. In the same context, Mudhsh & Almodfer (2017) obtained a validation error of 0.34% on the MADBase dataset by using an Alphanumeric VGG network inspired by the VGGNet (Simonyan & Zisserman, 2015) with dropout regularization and data augmentation but the error performance does not hold on the test set.

Torki et al. (2014) introduced AIA9K dataset and reported a classification error of 5.72% on the test set by using window-based descriptors with some common classifiers such as logistic regression, linear SVM, nonlinear SVM and artificial neural networks (ANN) classifiers. Younis (2017) tested a CNN architecture and obtained an error of 5.2%, he proposed a stacked CNN of three convolution layers followed by batch normalization, rectified linear units (ReLU) activation, dropout and two FC layers.

The AHCD dataset was introduced by El-Sawy, Loey & EL-Bakry (2017), they reported a classification error of 5.1% using a stacked CNN of two convolution layers, two pooling layers and two FC layers. Najadat, Alshboul & Alabed (2019) obtained a classification error of 2.8% by using a series CNN of four convolution layers activated by ReLU, two pooling layers and three FC layers. The state-of-the-art result for this dataset is a classification error of 1.58% obtained by Sousa (2018), it was achieved by ensemble averaging of four CNNs, two inspired by VGG16 and two written from scratch, with batch normalization and dropout regularization, to form 12 layers architecture called VGG12.

For benchmarking machine learning algorithms on tiny grayscale images other than Alphanumeric characters, Xiao, Rasul & Vollgraf (2017) introduced the Fashion-MNIST dataset to serve as a direct replacement for the original MNIST dataset and reported a classification test error of 10.3% using SVM. This dataset gained the attention of many researchers to test their approaches and better error of 3.65% was achieved by Zhong et al. (2017) in which a random erasing augmentation was used with wide residual networks (WRN) (Zagoruyko & Komodakis, 2016). The state-of-the-art performance for Fashion-MNIST is an error of 2.34% reported in Zeng et al. (2018) using a deep collaborative weight-based classification method based on VGG16. Recently, a modelling and optimization based method was used (Chou et al., 2019) to optimize the parameters for a multi-layer (16 layer) CNN reporting an error of 8.32% and 0.57% for Fashion-MNIST and MNIST respectively.

ALPR is a group of techniques that use CR modules to recognize vehicle’s LP number. Sometimes, it is also referred to as license plate detection and recognition (LPDR). ALPR is used in many real-life applications (Du et al., 2013) like electronic toll collection, traffic control, security, etc. The main challenges of detection and recognition of license plates are the variations in the plate types, environments, languages and fonts. Both CNN and traditional approaches are used to solve vehicle license plates recognition problems. Traditional approaches involve computer vision, image processing and pattern recognition algorithms for features such as color, edge and morphology (Xie et al., 2018). A typical ALPR system consists of three modules, plate detection, character segmentation and CR modules (Chang et al., 2004). This research focuses on CR techniques and compared them with the proposed CR approach. CR modules need an off-line training phase to train a classifier on each isolated character using a set of manually cropped character images (Bulan et al., 2017). Excessive operational time, cost and efforts must be considered when manual cropping of character images are needed to be collected and labeled for training and testing, and to overcome this, artificially generated synthetic license plates were proposed (Bulan, Kozitsky & Burry, 2015).

Additionally, very little research was done on multi-language LP character recognition, the reason is mostly due to the lack of multi-language LP datasets. Some recent researches were interested in introducing a global ALPR system. Asif et al. (2017) studied only LP detection module using a histogram-based approach, and a private dataset was used, which comprised of LPs from Hungary, America, Serbia, Pakistan, Italy, China, and UAE (Asif et al., 2017). VGG and LSTM were proposed for CR module in Dorbe et al. (2018) and the measured CR module accuracy was 96.7% where the test was done on LPs from Russia, Poland, Latvia, Belarus, Estonia, Germany, Lithuania, Finland and Sweden. Also, tiny YOLOv3 was used as a unified CR module for LPs from Greece, USA, Croatia, Taiwan, and South Korea (Henry, Ahn & Lee, 2020). Furthermore, several proposed methods interested in multi-language LPCR testing CR modules on each LP country’s dataset separately, without accumulating the characters into one dataset (Li, Wang & Shen, 2019; Yépez, Castro-Zunti & Ko, 2019; Asif et al., 2019). In addition, Selmi et al. (2020) proposed a mask R-CNN detector for character segmentation and recognition concerning Arabic and English LP characters from Tunisia and USA. Park, Yoon & Park (2019) concerned USA and Korean LPs describing the problem as multi-style detection. CNN shrinkage-based architecture was studied in Salemdeeb & Erturk (2020), utilizing the maximum number of convolutional layers that can be added. Salemdeeb & Erturk (2020) studied the LP detection and country classification problem for multinational and multi-language LPs from Turkey, Europe, USA, UAE and KSA, without studying CR problem. These researches studied LPs from 23 different countries where most of them use Latin characters to write the LP number, and totally five languages were concerned (English, Taiwanese, Korean, Chinese and Arabic). In Taiwan, Korea, China, UAE, Tunisia and KSA, the LP number is written using Latin characters, but the city information is coded using characters from that the country’s language.

In this paper, Arabic and Latin isolated characters are targeted to be recognized using a proposed full depth CNN (FDCNN) architecture in which the regions of interest are USA, EU and Middle East. To verify the performance of the proposed FDCNN, some isolated handwritten Arabic and Latin characters benchmarks such as MNIST, MADbase, AHCD, AIA9K datasets are also tested. Also, a new dataset named LP Arabic and Latin isolated characters (LPALIC) is introduced and tested. In addition, the recent FashionMNIST dataset is also tested to generalize the full depth feature extraction approach performance on tiny grayscale images. The proposed FDCNN approach closes the gap between software and hardware implementation since it provides low complexity and high performance. All the trained models and the LPALIC dataset (https://www.kaggle.com/dataset/b4697afbddab933081344d1bed3f7907f0b2b2522f637adf15a5fcea67af2145) are made publicly available online for research community and future tests.

The rest of this paper is organized as follows; “Datasets” introduces the structure of datasets used in this paper and also the new LPALIC dataset. In “Proposed Approach”, the proposed approach is described in details. “Experimental Results and Discussion” presents a series of experimental results and discussions. Finally, “Conclusion” summarizes the main points of the entire work as a conclusion.

Datasets

Datasets available in the literature

MNIST is a low-complexity data collection of handwritten digits to test supervised machine learning algorithms introduced by Lecun et al. (1998). It has grayscale images of size 28 × 28 pixels with 60,000 training digits and 10,000 test digits written by different persons. The digits are white and have black background, normalized to 20 × 20 pixels preserving the aspect ratio, and then centered at the center of mass of the 28 × 28 pixels grayscale images. The official site for the dataset and results are availabe by LeCun (http://yann.lecun.com/exdb/mnist/).

In MADbase, 700 Arabic native writers wrote ten digits 10 times and the images were collected as 70,000 binary images; 60,000 for training and 10,000 for testing, so that writers of training set and test set are exclusive. This dataset (http://datacenter.aucegypt.edu/shazeem) has the same format as MNIST to make veracity for comparisons between digits (used in Arabic and English languages) recognition approaches. Table 1 shows example digits of printed Latin, Arabic and handwritten Arabic characters used for numbers as declared in ISO/IEC 8859-6:1999.

Table 1 Printed and handwritten digits.

	

AHCD dataset (https://www.kaggle.com/mloey1/ahcd1) consists of 13,440 training images and 3,360 test images for 28 Arabic handwritten letters (classes) of size 32 × 32 pixels grayscale images. In AI9IK (http://www.eng.alexu.edu.eg/~mehussein/AIA9k/index.html) dataset, 62 female and 45 male Arabic native writers aged between 18 to 25 years old at the Faculty of Engineering at Alexandria University-Egypt were invited to write all the rabic letters three times to gather 8,988 letters of which 8,737 32 × 32 grayscale letter images were accepted after a verification process by eliminating cropping errors, writer mistakes and unclear letters. FasionMNIST dataset (github.com/zalandoresearch/fashion-mnist) has images of 70,000 unique products taken by professional photographers. The thumbnails (51 × 73) were then converted to 28 × 28 grayscale images by the conversion pipeline declared in Xiao, Rasul & Vollgraf (2017). It is composed of 60,000 training images and 10,000 test images of 10 class labels.

Table 2 gives a brief review on some publicly available related LP datasets for LPDR problem. The Zemris dataset is also called English LP in some references (Panahi & Gholampour, 2017).

Table 2 A review of publicly available ALPR datasets.

Dataset	Approach	Number of images	Accuracy (%)	Classifier	Character set	Purpose	
Zemris	Kraupner (2003)	510	86.2	SVM	No	LPDR	
UCSD	Dlagnekov (2005)	405	89.5	OCR	No	LPDR	
Snapshots	Martinsky (2007)	97	85	MLP	No	LPDR	
ARG	Fernández et al. (2011)	730	95.8	SVM	No	LPDR	
SSIG	Gonçalves, Menotti & Schwartz (2016)	2,000	95.8	SVM-OCR	Yes	LPDR	
ReId	Špaňhel et al. (2017)	77 k	96.5	CNN	No	LPR	
UFPR	Laroca et al. (2018)	4,500	78.33	CR-NET	Yes	LPDR	
CCPD	Xu et al. (2018)	250 k	95.2	RPnet	Yes	LPDR	

Novel license plate characters dataset

This research introduces a new multi-language LP chatacters dataset, involving both Latin and Arabic characters from LP images used in Turkey, USA, UAE, KSA and EU (Croatia, Greece, Czech, France, Germany, Serbia, Netherlands and Belgium). It is called LPALIC datase. In addition, some characters cropped from Brazil, India and other countries were added for just training to give features diversity. Furthermore, Some characters were collected from some public LP datasets, LP websites and our own camera pictures in Turkey taken in different weather conditions, places, blurring, distances, tilts and illuminations. These characters are real LP manually cropped characters without any filtering. For uniformity a size of 28 × 28 pixels of grayscale images was utilized.

The manually cropped characters were fed into the following conversion pipeline inspired from FashionMNIST (Xiao, Rasul & Vollgraf, 2017) which is similar to MNIST (Lecun et al., 1998),Resizing the longest edge of the image to 24 to save the aspect ratio.

Converting the image to 8-bit grayscale pixels image.

Negating the intensities of the image to get white character with black background.

Computing the center of mass of the pixels.

Translating the image to put center of mass at the center of the 28 × 28 grayscale image.

Some samples of the LPALIC dataset is visualized in Fig. 1 for Latin characters and in Fig. 2 for Arabic characters.

Figure 1 Samples of Latin characters in the LPALIC dataset.

Figure 2 Samples of Arabic characters in the LPALIC dataset.

Characters “0” and “O” are in the same class label so Latin characters have 35 (10 digits and 25 letters) class labels and Arabic characters have 27 (10 digits and 17 letters) class labels as LP as used in KSA. Table 3 illustrates the total number of Arabic and Latin characters included in LPALIC dataset.

Table 3 LPALIC dataset number of cropped characters per country.

Country	TR	EU	USA	UAE	Others	KSA	
Used characters	Latin	Latin	Latin	Latin & Arabic	Latin	Arabic	
Number of characters	60,000	32,776	7,384	3,003	17,613	50,000	
Total characters	120,776	50,000	

The Latin characters were collected from 11 countries (LPs have different background and font colors) while the Arabic characters were collected from only KSA (LPs have white background and black character). Choosing those countries is related to the availability of those LPs for public use.

Proposed approach

Stacked CNN architecture is simple, where each layer has a single input and a single output. For small size images, the key efficient simple deep learning architecture was LeNet-5 (Lecun et al., 1998), it consists of three convolutional, two pooling and one FC (Dense) layer. It was used and developed for the models in Cireşan et al. (2010) and in the main column of MCDNN in Ciregan, Meier & Schmidhuber (2012). Most of recent architectures are sparse structure of CNN such as models in GoogleNet (Szegedy et al., 2015), ResNet (He et al., 2016) and DensNet (Huang et al., 2017).

Proposed architecture

The core of the proposed model is the convolution block which is a convolutional layer followed by a batch normalization (BN) layer (Ioffe & Szegedy, 2015) and a non-linear activation ReLU layer (Krizhevsky, Sutskever & Hinton, 2012). This block is called standard convolutional layer in (Howard et al., 2017). The proposed convolutional layers have kernels of size 5 × 5 with a single stride. This kernel size showed a good feature extraction capability in LeNet-5 (Lecun et al., 1998) for small images as it covers 3.2% of the input image in every stride. However, the recent trends are to replace 5 × 5 with 2 layers of 3 × 3 kernels as in InceptionV3 (Szegedy et al., 2016). Figure 3 shows the architecture design of the proposed model.

Figure 3 Proposed FDCNN model architecture.

BN layer was recently introduced by Ioffe & Szegedy (2015). It normalizes the input by subtracting the mean of batch and dividing by the batch standard deviation then it scales and shifts the normalized input by learnable scale and shift, it reduces covariance shift, reduces overfitting, enables higher learning rates, regularizes the model and fulfills some of the same goals as Dropout layers. The first designers used BN layer in InceptionV3 are Szegedy et al. (2016).

For a mini-batch B = {x1, x2, . . ., xm} of size m, the mean μB and variance σB2 of B is computed and each input image in the mini-batch is normalized according to Eq. (1).

(1) x^i=(xi)−μBσB2+ε

where ε is a constant, x^i is the ith normalized image scaled by learnable scale parameter γ and shifted by learnable shift parameter β producing the ith normalized output image yi (Ioffe & Szegedy, 2015).

(2) yi=BNγ,β,(xi)=γx^i+β

Motivated by LeNet-5 convolution kernel 5 × 5, BN used in InceptionV3 and ReLU in Alexnet, the proposed model convolution block is built as in Fig. 4.

Figure 4 Proposed model convolution blocks.

The size of output feature map (FM) of each convolution block has lower size than the input feature map if no additional padding is applied. Eq. (3) describes the relation between input and output FM sizes (Goodfellow, Bengio & Courville, 2016).

(3) Wy=Wx−Wk+2PWs+1

where Wy is the width of the output, Wx is the width of the input, Wk is the width of the kernel, Ws is the width of the stride kernel and P is the number of padding pixels. For the height H, Eq. (3) can be used by replacing W with H. This reduction is called the shrinkage of convolution and it limits the number of convolutional layers that the network can include (Goodfellow, Bengio & Courville, 2016). The feature map shrinks from borders to the center as convolutional layers as added. Eventually, feature maps drop to 1 × 1 × channels (single neuron per channel) at which no more convolutional layers can be added. This is the concept of full depth used for designing the proposed architecture, Fig. 5 describes the full depth idea in FDCNN, where width and height shrink by 4 according to Eq. (3). In Fig. 5, each feature map is shrunk to a single value and this means that the features are convoluted into a single value resulting low number of parameters and high accuracy.

Figure 5 Full Depth concept of FDCNN.

The proposed FDCNN model composed basically of two stacked convolutional stages and one FC layer for 28 × 28 input images. Every stage has two convolution blocks and one max-pooling layer. It has a single input and a single output in all of its layers. Figure 3 shows the FDCNN architecture.

Parameter selection

In the proposed architecture, there are some parameters have to be selected, these parameters are kernel sizes of convolution, pooling layers kerenl sizes, the number of filters (channels) in convolution layers and strides. The kernel sizes are selected to be 5 × 5 for convolutional layers and 2 × 2 for pooling layers as described in architecture design in the previous Proposed Architecture section.

In literature, the trend for selecting the number of filters is to increase the number of filters as deep as the network goes (Krizhevsky, Sutskever & Hinton, 2012; Szegedy et al., 2015; Simonyan & Zisserman, 2015; He et al., 2016). Generally, the first convolutional layers learn simple features while deeper layers learn more abstract features. Selecting optimal parameters is based on heuristics or grid searches (Bengio, 2012). The rule of thumb to design a network from scratch is to start with 8–64 filters per layer and double the number of filters after each pooling layer (Simonyan & Zisserman, 2015) or after each convolutional layer (He et al., 2016). Recently, a new method was proposed to select the number of filters (Garg, Panda & Roy, 2018), an optimization of network structure in terms of both the number of layers and the number of filters per layer was done using principal component analysis on trained network with a single shot of analysis. A 9X reduction in the number of operations and up to 3X reduction in the number of parameters with less than 1% drop in accuracy is achieved upon training on the same task. In context, a modeling and optimization method (MAOM) was proposed in Chou et al. (2019) to optimize CNN parameters by integrating uniform experimental design (UED) and multiple regression (MR), but the rule of thumb for doubling the number of filters was also applied.

One of the contributions of this research is to select the number of channels that achieves full depth. Number of filters may also be called the number of kernels, number of layer channels or layers width. The number of filters is selected to be as the same as the number of shrinking pixels in each layer from bottom to the top. Table 4 shows the shrinkage of the proposed model. From the fact that the network goes more deeper the following selection is made:

Table 4 Shrinkage process in 28 × 28 architecture.

Layer	Shrinking pixels	Width	
Conv1	282 − 242 = 208	64	
Conv2	242 − 202 = 176	128	
Max-Pooling 1	—	128	
Conv3	102 − 62 = 64	176	
Conv4	62 − 22 = 32	208	
Max-Pooling 2	—	208	

The width of 4th convolutional layer is 208 (the 1st layer shrinkage).

The width of 3rd is 176 (the the 2nd layer shrinkage).

The max-pooling will make a loss of half in FM dimensions so the next layers shrinkage pixels will be doubled.

The width of 2nd is 128 (the double of the 3rd layer shrinkage).

The width of 1nd is 64 (the double of the 4th layer shrinkage).

The same parameter selection method can be applied to 32 × 32 input architecture as described in Table 5 to reach full depth features (single value feature) as shown in Fig. 5B.

Table 5 Shrinkage process in 32 × 32 architecture.

Layer	Shrinking pixels	Width	
Conv1	322 − 282 = 240	64	
Conv2	282 − 242 = 208	128	
Conv3	242 − 202 = 176	176	
Max-pooling 1	—	176	
Conv4	102 − 62 = 64	208	
Conv5	62 − 22 = 32	240	
Max-pooling 2	—	240	

Table 6 shows the number of learnable parameters and feature memory usage for the proposed model. Memory usage is multiplied by 4 as each pixel is stored as 4-byte single float number. For 32 × 32 input images just another convolutional block can be added before the first convolution block in FDCNN and the width of last convolutional layer will be 322 − 282 = 240 to get full depth of shrinkage. This layer of course affects the total number of model parameters and FM memory usage to be 2.94 M and 2.51 MB respectively.

Table 6 Proposed model’s memory usage and learnable parameters.

Layer	Features memory	Learnable parameters	
Input	28 × 28 × 1	3,136	0	0	
Conv1	24 × 24 × 64	147,456	(5 × 5) × 64 + 64 =	1,664	
BN+ReLU	24 × 24 × 64 × 2	294,912	4 × 64 =	256	
Conv2	20 × 20 × 128	204,800	5 × 5 × 64 × 128 + 128 =	204,928	
BN+ReLU	20 × 20 × 128 × 2	409,600	4 × 128 =	512	
Max-pooling1	10 × 10 × 128	51,200	0	0	
Conv3	6 × 6 × 176	25,344	5 × 5 × 128 × 176 + 176 =	563,376	
BN+ReLU	6 × 6 × 176 × 2	50,688	4 × 176 =	704	
Conv4	2 × 2 × 208	3,328	5 × 5 × 176 × 208 + 208 =	915,408	
BN+ReLU	2 × 2 × 208 × 2	6,656	4 × 208 =	832	
Max-pooling2	1 × 1 × 208	832	0	0	
FC	1 × 1 × 10	40	208 × 10 + 10 =	2,090	
	Total memory	1,197,992 byte	Total parameters	1,689,770	

In general, DNNs give weights for all input features (neurons) to produce the output neurons, but this needs a huge number of parameters. Instead, CNNs convolve the adjacent neurons by the convolution kernel size to produce the output neurons. In the literature, the state-of-the-art architectures had high number of learnable parameters at the last FC layers. For example, VGG16 has totally 136 M parameters, and after the last pooling layer the first FC layer has 102 M parameters, which means more than 75% of the architecture parameters (just in one layer). AlexNet has totally 62 M parameters, and the first FC layer has 37.75 M parameters, which means more than 60% of the architecture parameters. In (Hirata & Takahashi, 2020), the proposed architecture has 28.68 M parameters, and the first FC layer has 3.68 M parameters after majority voting from ten divisions. But, by using the full depth concept to reduce FM to 1 × 1 size after the last pooling layer, FDCNN has just 2,090 parameters from totally 1.6 M parameters as seen in Table 6. The full depth concept of reducing the feature maps size to one neuron has decreased the total number of learnable parameters which make FDCNN simple and fast.

Training process

Deep learning training algorithms were well explained in Goodfellow, Bengio & Courville (2016). The proposed model is trained using stochastic gradient descent with momentum (SGDM) with custom parameters chosen after many trails. initial learning rate (LR) of 0.025, mini-batch size equals to the number of training instances divided by number of batches needed to complete one epoch, LR drop factor by half every 2 epochs, 10 epochs, 0.95 momentum and the training set is shuffled every epoch. However, those training parameters are not used for all datasets since the number of images is not constant in all of them.

After getting the first results, The model parameters are tuned by training again on ADAM with larger mini-batch size and very small LR started by 1 × 10−5, then multiplying the batch size by 2 and LR by 1/2 every 10 epochs as long as the test error has improvement.

Experimental results and discussion

All of training and testing are made on MATLAB2018 platform with GeForce 1,060 (6 GB shared memory GPU). The main goal of this research is to design a CNN to recognize multi-language characters of license plates but to generalize and verify the designed architecture several tests on handwritten character recognition benchmarks are done (verification process). The proposed approach showed very promising results. Table 7 summarizes the results obtained on MNIST dataset.

Table 7 Test results of FDCNN on MNIST.

Architecture	Type	Number of layers	Error (%)	
Cireşan et al. (2010)	Stacked	15	0.35	
Ciresan et al. (2011)	Sparse	35	0.27	
Ciregan, Meier & Schmidhuber (2012)	Sparse	245	0.23	
Moradi, Berangi & Minaei (2019)	Sparse	70	0.28	
Kowsari et al. (2018)	Sparse	—	0.18	
Assiri (2019)	Stacked	15	0.17	
Hirata & Takahashi (2020)	Sparse	28	0.16	
Byerly, Kalganova & Dear (2020)	Sparse	21	0.16	
Proposed	Stacked	12	0.28	

It is clear that stacked CNN has not outperformed the error of 0.35% in the literature for MNIST but the approach used in (Assiri, 2019) obtained 0.17%. The proposed FDCNN performance approximately reached close to the performance of five committees CNN of (Ciresan et al., 2011). FDCNN do as the same performance as (Moradi, Berangi & Minaei, 2019) which it is a sparse design that uses Residual blocks and Inception blocks as described in the literature. However, the architecture in (Assiri, 2019) has 15 layers with 13.12 M parameters, the results were obtained utilizing data augmentations, different training processes and Dropout layers before and after each pooling layer with different settings. FDCNN has less parameters and layers and showed good results on MNIST.

On the other hand, the proposed approach is tested on MADbase, AHCD and AI9IK datasets for Arabic character recognition benchmarks to verify FDCNN and to generalize using it in Arabic ALPR systems. Table 8 describes the classification error regarding the stat-of-the-art on such datasets.

Table 8 Arabic character recognition benchmarks state-of-the-art and proposed approach test errors.

Dataset	Architecture	Type	Layers	Parameters	Error	
MADbase
28 × 28	RBF SVM
Abdleazeem & El-Sherif (2008)	Linear	—	—	0.52%	
LeNet5
El-Sawy, EL-Bakry & Loey (2017)	Stacked	7	51 K	12%	
Alphanumeric VGG
Mudhsh & Almodfer (2017)	Stacked	17	2.1 M	0.34% validation	
VGG12 REGU
Sousa (2018)	Average of 4 stacked CNN	66	18.56 M	0.48%	
Proposed	Stacked	12	1.69 M	0.34%	
AHCD
32 × 32	CNN
El-Sawy, Loey & EL-Bakry (2017)	Stacked	7	1.8 M	5.1%	
CNN
Younis (2017)	Stacked	6	200 K	2.4%	
VGG12 REGU
Sousa (2018)	Average of 4 stacked CNN	66	18.56 M	1.58%	
CNN
Najadat, Alshboul & Alabed (2019)	Stacked	10	Not mentioned	2.8%	
Proposed	Stacked	13	2.94 M	1.39%	
AI9IK
32 × 32	RBF SVM
Torki et al. (2014)	Linear	—	—	5.72%	
CNN
Younis (2017)	Stacked	6	200 K	5.2%	
Proposed	Stacked	13	2.94 M	3.27%	

As seen in Table 8, for MADbase dataset, most of the tested approaches were based on VGG architecture. Alphanumeric VGG (Mudhsh & Almodfer, 2017) reported a validation error of 0.34% that did not hold on the test set while FDCNN obtained 0.15% validation error and 0.34% test error. The proposed approach outperformed Arabic character recognition benchmarks state-of-the-arts for both digits and letters used in Arabic language with less number of layers and learnable parameters. It has succeed this verification process on these datasets too.

In Table 8, input layer is included in the determination of the number layers (Lecun et al., 1998) for all architectures and ReLU layer is not considered as a layer but BN is considered as a layer. Sousa (2018) considered convolution, pooling and FC layers when the number of layers was declared but four trained CNNs were used with softmax averaging, this is why the number of layers and learnable parameters are high. Najadat, Alshboul & Alabed (2019) did not declare the most of network parameters like kernel size in every convolution layer and they changed many parameters to enhance the model. In Younis (2017), 28 × 28 input images were used and no pooling layers were included.

On the other hand, and in the same verification process, the proposed approach is tested on FashionMNIST benchmark too to generalize using it over grayscale tiny images. As shown in Table 9, the proposed approach outperformed the stacked CNN architectures and reached near DENSER network in Assunção et al. (2018) and EnsNet in Hirata & Takahashi (2020) with less layers and parameters but with a good performance. It can be said that FDCNN has a very good verification performance on FashionMNIST dataset. FDCNN outperformed (Byerly, Kalganova & Dear, 2020) results on Fashion-MNIST benchmark while (Byerly, Kalganova & Dear, 2020) outperformed FDCNN on MNIST.

Table 9 Test results of FDCNN on FashionMNIST.

Architecture	Type	Layers	Parameters	Error (%)	
SVM (Xiao, Rasul & Vollgraf, 2017)	Linear	—	—	10.3	
DENSER (Assunção et al., 2018)	Sparse	—	—	4.7	
WRN (Zhong et al., 2017)	Sparse	28	36.5 M	3.65	
VGG16 (Zeng et al., 2018)	Sparse	16	138 M	2.34	
CNN (Chou et al., 2019)	Stacked	16	0.44 M	8.32	
BRCNN (Byerly, Kalganova & Dear, 2020)	Sparse	16	1.51 M	6.34	
EnsNet (Hirata & Takahashi, 2020)	Sparse	28	28.67 M	4.7	
Proposed	Stacked	12	1.69 M	5.00	

Furthermore, FDCNN is tested also on Arabic LP characters from KSA. It could classify the test set with error of 0.46%. It outperformed the the recognition error results of 1.78% in Khaled, Rached & Hasan (2010). FDCNN has successfully verified on KSA Arabic LP characters dataset.

In this research and for more verification, FDCNN performance is also tested on both common publicly available LP benchmark characters and the new LPALIC dataset. Table 10 shows the promising results on LP benchmarks. FDCNN outperformed the stat-of-art results on common LP datasets for isolated character recognition problem. For Zemris and ReId datasets the proposed FDCNN was trained on LPALIC dataset and tested on all characters in both test sets. It is clear that FDCNN has efficiently verified on common LP benchmarks.

Table 10 Recognition error of proposed architecture on LP benchmarks datasets.

Architecture	Dataset	Layers	Parameters	Error (%)	
SVM (Panahi & Gholampour, 2017)	Zemris	—	—	3	
LCR-Alexnet (Meng et al., 2018)	12	>2.33 M	2.7	
Proposed	12	1.69 M	0.979	
OCR (Dlagnekov, 2005)	UCSD	—	—	10.5	
Proposed	12	1.69 M	1.51	
MLP (Martinsky, 2007)	Snapshots	—	—	15	
Proposed	12	1.69 M	0.42	
CNN (Špaňhel et al., 2017)	ReID	12	17 M	3.5	
DenseNet169 (Zhu et al., 2019)	169	>15.3 M	6.35	
Proposed	12	1.69 M	1.09	
CNN (Laroca et al., 2018)	ssUFPR	26	43.1 M	35.1	
Proposed trained just on UFPR	12	1.69 M	4.29	
Proposed trained on LPALIC	12	1.69 M	2.03	
Line processing algorithm (Khaled, Rached & Hasan, 2010)	KSA	—	—	1.78	
Proposed	12	1.69 M	0.46	
FDCNN	LPALIC	12	1.69 M	0.97	

For more analysis, another test is made on the introduced LPALIC dataset to analyze the recognition error on characters per country. Table 11 describes the results. As seen in Table 11, the highest error is in classifying USA LP characters because it has more colors, drawings and shapes other than characters and also there is a small number of instances in the characters dataset. However, a very high recognition accuracy is achieved on Turkey and EU since they have the same standard and style for LPs. In Turkey, 10 digits and 23 letter is used since letters like Q, W and X are not valid in Turkish language. Additionally, FDCNN could classify Arabic LP characters with very low error. UAE characters set has a small number of cropped characters that is why it is tested just by FDCNN trained on other countries character sets.

Table 11 Test recognition error per country characters with different training instances.

Characters set	Number of instances Train/test	Manual split	Trained on other countries	Random 80/20% split average error (%)	Random 70/10/20% split average error (%)	
TR	48,748/11,755	2.67%	1.82%	0.97	0.99	
EU	23,299/9,477	2.30%	1.07%	1.03	0.80	
USA	5,960/1,424	10.88%	3.51%	1.96	1.79	
UAE	1,279/1,724	—	1.51%	0.9	1.08	
All Latin characters	96,899/24,380	2.08%	—	0.97	1.06	
KSA	46,981/3,018	0.43%	—	0.26	0.30	

To make robust tests, the characters were split manually and randomly as seen in Table 11. In manual split, the most difficult characters were put in the test set and the others in the training set while in random split 80% were split for training and the rest of them for testing. As described in Table 3, the number of characters per country is not equal, which resulted various recognition accuracies in Table 11. Since the number of UAE characters is not large enough to train FDCNN, Latin characters from other countries were used for training but the test was done only on UAE test set. FDCNN could learn features that give good average accuracy. In fact, the Latin characters in LPALIC have various background and foreground colors which make the classification more challenging than Arabic characters set, but FDCNN shows a promising recognition results on both and also on handwritten characters as well.

In the manual split in Table 11, the country’s characters training and testing sets were used to train and test FDCNN. In trained on other countries, the FDCNN was trained on both the country’s characters training set and other countries characters but tested only on that country’s test set. In the random 80/20 split, the country’s characters were split randomly into training and testing sets, and FDCNN was trained on both the split country’s characters training set and other countries characters but tested only on that split country’s test set, a lot of random split tests were done and the average errors were reported in the table.

Furthermore, in Table 11, validation sets were also used to guarantee in a sufficiently clear way that the results were not optimized specifically for those test sets. 70% of the dataset is randomly split for training, 10% for validation and 20% for testing. The training hyperparameters were optimized on a validation set, and the best parameters for the validation set were then be used to calculate the error on the test set. Those different test analyses were done to validate and evaluate the results and reduce the overfitting problem. In fact, the Latin characters in LPALIC have various background and foreground colors which make the classification more challenging than Arabic characters set, but FDCNN shows a promising recognition results on both and also on handwritten characters as well.

Conclusion

This research focused on deep learning technique of CNNs to recognize multi-language LP characters for both Latin and Arabic characters used in vehicle LPs. A new approach is proposed, analyzed and tested on Latin and Arabic CR benchmarks for both LP and handwritten characters recognition. The proposed approach consists of proposing FDCNN architecture, FDCNN parameter selection and training process. The proposed full depth and width selection ideas are very efficient in extracting features from tiny grayscale images. The complexity of FDCNN is also analyzed in terms of number of learnable parameters and feature maps memory usage.The full depth concept of reducing the feature maps size to one neuron has decreased the total number of learnable parameters while achieving very good results. Implementation of FDCNN approach is simple and can be used in real time applications worked on small devices like mobiles, tablets and some embedded systems. Very promising results were achieved on some common benchmarks like MNIST, FashionMNIST, MADbase, AIA9K, AHCD, Zemris, ReId, UFPR and the newly introduced LPALIC dataset. FDCNN performance is verified and compared to the state-of-the-art results in the literature. A new real LPs cropped characters dataset is also introduced. It is the largest dataset for LP characters in Turkey and KSA. More tests can be done on FDCNN for future work to be the core of CNN processor. Also, more experiments can be conducted to hybrid FDCNN with some common blocks like residual and inception blocks. Additionally, the proposed full depth approach may be applied to other stacked CNNs like Alexnet and VGG networks.

Supplemental Information

Supplemental Information 1 Zemris set train test code.

Open the file using MATLAB.

Click here for additional data file.

Supplemental Information 2 Zemris Set.

Open the file using MATLAB.

Click here for additional data file.

Supplemental Information 3 UFPR dataset train and test code.

Open the file using MATLAB.

Click here for additional data file.

Supplemental Information 4 UFPR dataset train on LPALIC dataset and test on UFPR test set.

Open the file using MATLAB.

Click here for additional data file.

Supplemental Information 5 UFPR characters set.

Open the file using MATLAB.

Click here for additional data file.

Supplemental Information 6 UFPR test results if FDCNN trained on LPALIC dataset.

Open the file using MATLAB.

Click here for additional data file.

Supplemental Information 7 UFPR test if FDCNN trained on UFPR training ser.

Open the file using MATLAB.

Click here for additional data file.

Supplemental Information 8 FDCNN trained model tested on Zemris set.

Open the file using MATLAB.

Click here for additional data file.

Supplemental Information 9 FDCNN trained on LPALIC and tested on UCSD set.

Open the file using MATLAB.

Click here for additional data file.

Supplemental Information 10 UCSD character set.

Open the file using MATLAB.

Click here for additional data file.

Supplemental Information 11 FDCNN trained on LPALIC and test result on UCSD.

Open the file using MATLAB.

Click here for additional data file.

Supplemental Information 12 FDCNN training on LPALIC and Testing on Snapshots set code.

Open the file using MATLAB.

Click here for additional data file.

Supplemental Information 13 Snapshots characters set.

Open the file using MATLAB.

Click here for additional data file.

Supplemental Information 14 FDCNN training on LPALIC and testing on RIed_HDR code.

Open the file using MATLAB.

Click here for additional data file.

Supplemental Information 15 RIed HDR characters set.

Open the file using MATLAB.

Click here for additional data file.

Supplemental Information 16 FDCNN trained on LPALIC and tested on Snapshots set result.

Open the file using MATLAB.

Click here for additional data file.

Supplemental Information 17 FDCNN trained on LPALIC and tested on HDR set result.

Open the file using MATLAB.

Click here for additional data file.

Supplemental Information 18 FDCNN trained on random 80% of USA set and tested on 20% of USA set code.

Open the file using MATLAB.

Click here for additional data file.

Supplemental Information 19 FDCNN training on manula split of USA code.

Open the file using MATLAB.

Click here for additional data file.

Supplemental Information 20 USA characters set.

Open the file using MATLAB.

Click here for additional data file.

Supplemental Information 21 FDCNN trained on random split of USA set result.

Open the file using MATLAB.

Click here for additional data file.

Supplemental Information 22 FDCNN trained on random split of USA set result.

Open the file using MATLAB.

Click here for additional data file.

Supplemental Information 23 FDCNN trained on random split of TR set result.

Open the file using MATLAB.

Click here for additional data file.

Supplemental Information 24 FDCNN trained on random split of USA set result.

Open the file using MATLAB.

Click here for additional data file.

Supplemental Information 25 TR characters set.

Open the file using MATLAB.

Click here for additional data file.

Supplemental Information 26 FDCNN UAE set manual split Training Testing.

Open the file using MATLAB.

Click here for additional data file.

Supplemental Information 27 LPALIC excluded TR set for training FDCNN and test on TR test set results.

Open the file using MATLAB.

Click here for additional data file.

Supplemental Information 28 FDCNN trained on 80% random split of KSA and testing 20% code.

Open the file using MATLAB.

Click here for additional data file.

Supplemental Information 29 FDCNN trained on 80% Random split and tested on 20% of UAE set.

Open the file using MATLAB.

Click here for additional data file.

Supplemental Information 30 FDCNN training on random 80% of KSA and testing on 20% code.

Open the file using MATLAB.

Click here for additional data file.

Supplemental Information 31 UAE character set.

Open the file using MATLAB.

Click here for additional data file.

Supplemental Information 32 FDCNN training on manual split of KSA code.

Open the file using MATLAB.

Click here for additional data file.

Supplemental Information 33 FDCNN trained on LPALIC dataset and tested on UAE set.

Open the file using MATLAB.

Click here for additional data file.

Supplemental Information 34 FDCNN training on 80% of LPALIC and testing on 20 TR code.

Open the file using MATLAB.

Click here for additional data file.

Supplemental Information 35 FDCNN trained on manual split of KSA result.

Open the file using MATLAB.

Click here for additional data file.

Supplemental Information 36 FDCNN traing on TR set manual split code.

Open the file using MATLAB.

Click here for additional data file.

Supplemental Information 37 FDCNN trained on random split of KSA result.

Open the file using MATLAB.

Click here for additional data file.

Supplemental Information 38 KSA manually split characters set.

Open the file using MATLAB.

Click here for additional data file.

Supplemental Information 39 KSA characters set.

Open the file using MATLAB.

Click here for additional data file.

Supplemental Information 40 FDCNN manual split of EU set training and testing code.

Open the file using MATLAB.

Click here for additional data file.

Supplemental Information 41 FDCNN training and testing on random split 80% and 20% code.

Open the file using MATLAB.

Click here for additional data file.

Supplemental Information 42 FDCNN trained on manual split of EU set result.

Open the file using MATLAB.

Click here for additional data file.

Supplemental Information 43 FDCNN trained on manual split of LPALIC result.

Open the file using MATLAB.

Click here for additional data file.

Supplemental Information 44 EU characters set.

Open the file using MATLAB.

Click here for additional data file.

Supplemental Information 45 FDCNN training and testing on random 80% and 20% of LPALIC code.

Open the file using MATLAB.

Click here for additional data file.

Supplemental Information 46 FDCNN training and testing on manual split of LPALIC code.

Open the file using MATLAB.

Click here for additional data file.

Supplemental Information 47 FDCNN trained on random 80% and 20% of LPALIC result.

Open the file using MATLAB.

Click here for additional data file.

Supplemental Information 48 FDCNN trained on manual split of LPALIC and test result.

Open the file using MATLAB.

Click here for additional data file.

Supplemental Information 49 FDCNN training and testing on MNIST code.

Open the file using MATLAB.

Click here for additional data file.

Supplemental Information 50 FDCNN training on MNIST code.

Open the file using MATLAB.

Click here for additional data file.

Supplemental Information 51 MNIST handwritten character set.

Open the file using MATLAB.

Click here for additional data file.

Supplemental Information 52 FDCNN trained on 5000 of MNIST and tested on MNSIT test set result.

Open the file using MATLAB.

Click here for additional data file.

Supplemental Information 53 FDCNN testing on MAHDB code.

Open the file using MATLAB.

Click here for additional data file.

Supplemental Information 54 FDCNN training on MAHDB code.

This is a ".m" which can be opened by the MATLAB program.

Click here for additional data file.

Supplemental Information 55 MAHDB Arabic handwritten digits characters set.

Open the file using MATLAB.

Click here for additional data file.

Supplemental Information 56 FDCNN trained and tested on MAHDB result.

Open the file using MATLAB.

Click here for additional data file.

Supplemental Information 57 FDCNN testing on ArabicHC code.

Open the file using MATLAB.

Click here for additional data file.

Supplemental Information 58 FDCNN training on AHCD code.

Open the file using MATLAB.

Click here for additional data file.

Supplemental Information 59 AHCD Arabic handwritten letters set.

Open the file using MATLAB.

Click here for additional data file.

Supplemental Information 60 FDCNN trained and tested on AHCD result.

Open the file using MATLAB.

Click here for additional data file.

Supplemental Information 61 FDCNN testing on AIA9K code.

Open the file using MATLAB.

Click here for additional data file.

Supplemental Information 62 FDCNN training on AIA9K code.

Open the file using MATLAB.

Click here for additional data file.

Supplemental Information 63 AIA9K Arabic handwritten letters dataset.

Open the file using MATLAB.

Click here for additional data file.

Supplemental Information 64 FDCNN trained and tested on AIA9K result.

Open the file using MATLAB.

Click here for additional data file.

Supplemental Information 65 FDCNN testing on fashionMNIST code.

Open the file using MATLAB.

Click here for additional data file.

Supplemental Information 66 FDCNN training on fashionMNIST code.

Open the file using MATLAB.

Click here for additional data file.

Supplemental Information 67 FDCNN trained and tested on fashionMNIST result.

Open the file using MATLAB.

Click here for additional data file.

Supplemental Information 68 fashionMNIST images dataset.

Open the file using MATLAB.

Click here for additional data file.

Additional Information and Declarations

Competing Interests

Author Contributions

Data Availability

The authors declare that they have no competing interests.

Mohammed Salemdeeb conceived and designed the experiments, performed the experiments, analyzed the data, performed the computation work, prepared figures and/or tables, authored or reviewed drafts of the paper, and approved the final draft.

Sarp Ertürk analyzed the data, authored or reviewed drafts of the paper, and approved the final draft.

The following information was supplied regarding data availability:

The trained models, codes, tests and datasets are available at Kaggle: SalemDeeb, M. (2020). “Multi-Language LP Character Recognition

Full Depth Convolutional Neural Network”. Kaggle. Dataset. https://www.kaggle.com/dataset/b4697afbddab933081344d1bed3f7907f0b2b2522f637adf15a5fcea67af2145.

All the files can be accessed using MATLAB.

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
