# Peer review of "Full depth CNN classifier for handwritten and license plate characters recognition"

_PeerJ Computer Science, doi:10.7717/peerj-cs.576_

## Round 0.1 · original submission · Major Revisions

The manuscript needs for a thorough revision. Thank you.

Reviewer 1 ·

Basic reporting

The paper presents a new approach to detect LP by stacking two CNN networks. It manages to obtain a state if the art results while using a small network in terms of the number of parameters.

Experimental design

The experimental Section is clear and well-designed.

Validity of the findings

The novelty of the approach is questionable, but the finding is interesting.

Additional comments

The paper presents another approach for LP recognition. The paper is clear, but there are several points that need further classification.
1. VGG seems to obtain the same performance, you should explain why you need another network
2. You need to explain why you have different results for the various datasets on average (table 11)
3. What is the point of including fashionMNIST
4. There is no Latin digits, the one you call Latin are Arabic and the one you call Arabic are Hindi

Reviewer 2 ·

Basic reporting

Some revisions are necessary on the presentation of the text itself. In my general comments to the authors, I’ve pointed out some grammar errors, typos and unclear passages.
This seems to warrant a full revision of the text to ensure the promising results of the paper are not obfuscated to readers.

Experimental design

The manuscript describes a classification architecture for license plates from multiple countries, which is also tested in common character and digit recognition datasets. Additionally it is also tested in FashionMNIST, a popular small grayscale image dataset. To my understanding, the strongest contributions of the paper are the proposed Full Depth CNN (FDCNN) model, and the proposal of a new license plate dataset called LPALIC. They seem to successfully address the knowledge gaps identified by the authors.

Regarding the FDCNN model, promising results are reported for all the studied datasets, including state of the art results for MNIST when only stacked CNNs are considered, according to the authors. I believe that the methodology description and results must be entirely reproducible, especially since state of the art results in widely used benchmarks are involved.

Because of this, I believe the paper needs revisions to allow for easier reproduction and verification of results, and also needs more justifications on certain choices in the methodology, as explained in my comments further below.

In this current form of the manuscript, readers might not understand why many aspects of this FDCNN architecture were chosen, or how they were tested and validated. The lack of the use of a validation set, for instance, should be clarified, as directly optimizing a model on the test set could potentially introduce biases. This is true even if the justifications are of an empirical nature. I believe the authors should be able to revise the manuscript to make all of these points clearer and further highlight and solidify their already strong results, before publication.

Validity of the findings

All results are well described and data and codes are provided, but the points mentioned about experimental design and basic reporting must be addressed to ensure the validity of the results.
I believe once those matters are addressed and clarified the validity of the findings will be more easily asserted.

Additional comments

Line 36 - It would be interesting to additionally address or discuss these recent works, which also reportedly provide state of the art accuracies for the same task of classifying MNIST:

Hirata and Takahashi; Ensemble Learning In CNN Augmented with Fully Connected Subnetworks
Byerly et al.; A Branching and Merging Convolutional Network with Homogeneous Filter Capsules
Assiri; Stochastic Optimization of Plain Convolutional Neural Networks with Simple methods
Kowsari et al.; RMDL: Random Multimodel Deep Learning for Classification

Line 84 – method was used

Line 100 – very little research was done

Line 100 – This paragraph starts with the idea that there are very few multi-language license plate datasets, but then cites works about license plate classification from what seems to be a considerable variety of countries and alphabet types. Please consider clarifying this paragraph by stating more explicitly how many datasets exist among the cited works, and all the different languages and character types used. This can create a clearer picture for the reader, and help to further highlight the new contributions of the paper.

Lines 101 to 104 - The grammar in this sentence is unclear. Please restructure.

Line 117 – Please revise the usage of “concerned” in this sentence.

Line 117 – Since the main contribution of the paper revolves around FDCNNs, it is important that the reader gets a clear understanding of other types of CNNs, what makes them different, and why the changes included in FDCNN are necessary. I feel that this information is currently lacking in the paper, and it’s not entirely clear how the authors justified pursuing this particular approach or why it’s necessary, or important, or better, to reduce featuremaps to 1x1 size before classification. Please include a more detailed discussion/justification.

Line 119 – Please restructure the sentence so FDCNN is not repeated twice in a row.

Line 158 – It would greatly help readers to have a table or something similar where all countries studied are listed, and what language/character types are used in license plates in that particular country.

Figures 1 and 2 – These figures show some interesting differences between datasets. Namely, Arabic alphabet LPs seem to vary much less in color. Could this possibly affect classification accuracies? There seems to be a marked imbalance in the number of Latin and Arabic images in the dataset. Would this warrant using additional metrics, besides accuracy, that are more sensitive to these imbalances? Please discuss this in the manuscript.

Line 187 – Please avoid qualitative/subjective qualifiers such as “modest”, used here. There are other examples of such adjectives in the manuscript. If necessary, compare directly to other values used in the literature, by choosing appropriate metrics for the comparison.

Line 205 – shrink

Line 250 – See comment about line 187. The same applies to the usage of “modest” in this line.

Line 251 – Please clarify the meaning of “needed iterations”. How is this defined?

Line 251 – Please specify batch size and momentum used. The supplemental files seem to show the minibatch used had a size of 120. They also mention a LearnRateDropFactor of 0.9, whereas the paper seems to mention one of 0.5 . It also seems that the standard momentum for Matlab’s ‘sgdm’ function is used, but this value is not stated explicitly in the paper. It would be interesting to add it.
Since one of the results of the paper is a demonstrable improvement upon the state of the art of stacked CNNs, it would be interesting, for reproducibility sake, to include these hyper-parameters in the methodology description.

Line 251- Additionally, it would be interesting to discuss how these parameters were chosen. If there were preliminary tests, or heuristics, how did other attempts affect the results, and by how much?
It would seem that no validation set was used, so how were metrics chosen? Was the model adjusted to get the best result directly on the test set, without using a separate validation set? Couldn’t this bias the models to adjust particularly well only to the test set?

Table 11 – The UAE test set is larger than the training set. I feel this decision should be justified in the paragraph above the table. I could not see a justification for it as is.

---

## Round 0.2 · Minor Revisions

Reviewers have now commented on the paper. Based on their suggestions, I'm happy to provide you a decision: minor revision.

Reviewer 3 ·

Basic reporting

A deeper error analysis is required.

Experimental design

The authors need to discuss the evaluation procedure.

Validity of the findings

The authors can discuss the results for other performance metrics.

Additional comments

The literature review needs to be updated with some of the recent works. Discussing the following works would make the manuscript richer for the readers:

DevNet: An Efficient CNN Architecture for Handwritten Devanagari Character Recognition. Int. J. Pattern Recognit. Artif. Intell. 34(12): 2052009:1-2052009:20 (2020)
Character recognition based on non-linear multi-projection profiles measure. Frontiers Comput. Sci. 9(5): 678-690 (2015)
Relative Positioning of Stroke-Based Clustering: a New Approach to Online Handwritten Devanagari Character Recognition. Int. J. Image Graph. 12(2) (2012)
Artistic Multi-character Script Identification Using Iterative Isotropic Dilation Algorithm. RTIP2R (3) 2018: 49-62
Character Recognition Based on DTW-Radon. ICDAR 2011: 264-268
Spatial Similarity Based Stroke Number and Order Free Clustering. ICFHR 2010: 652-657
Dtw-Radon-Based Shape Descriptor for Pattern Recognition. Int. J. Pattern Recognit. Artif. Intell. 27(3) (2013)

·

Basic reporting

A new approach to LP identification by stacking two CNN networks is discussed in the paper. The core of the convolution block, which is the convolution Layer is followed by batch normalization and a non-Linear activation layer.

Experimental design

There is a well-designed experimental section. The manuscript's strongest achievements are the suggested Full Depth CNN (FDCNN) model and the recommendation for a new license plate data set called LPALIC. The strength of the paper is the selection of parameters, filters, and the process of training. The manuscript also gives an overview of FDCNN with respect to the use of memory.
The drawback of the manuscript is that the results obtained during the testing process on the test dataset are missing and no validation dataset has been used, so it is unclear that how fine-tuning of the model hyperparameters has been performed and the problem of overfitting has been addressed. I think the writers should be able to update the manuscript before publishing to make all these points clear and further illustrate and solidify their already good findings

Validity of the findings

All outcomes are well defined and codes are given, but to ensure the validity of the data, the points listed must be answered. I assume that the relevance of the results can be more readily asserted if the issues are discussed and explained.

Additional comments

The paper is clear, but there are few points that require Clarification
1. Explains the methodology used in order to correctly recognize Arabic zero numbers and letters written in continuous style.
2.Discuss the criteria based on which a character is labeled as difficult /easy as specified in line number
342.
3.You should clarify, why one should go for the proposed FDCNN model, when  Assiri,2019 stacked method has outperformed.

---

## Round 0.3 · Minor Revisions

Authors are requested a few more suggestions. If you are able to revise accordingly, I would be happy to accept the paper. Thank you.

Reviewer 2 ·

Basic reporting

no comment

Experimental design

The authors still only use a training set and a test set, with no validation set. The text doesn't guarantee in a sufficiently clear way that the results weren't optimized specifically for this test set, which could mean the data is biased to fit this specific test set.

The training hyperparameters should be optimized on a validation set, and the best parameters for the validation set should then be used to calculate the test metrics.

Validity of the findings

The previous comment on validation sets must be clearly addressed by the authors. It is not the same thing to use a validation set and then a test set, and to optimize directly for the test set performance (which seems to be the case here).

Furthermore, always prefer to state where the performance is calculated. If the accuracy was computed on the test set, mention it as "test accuracy". This must be abundantly clear to the reader.

Additional comments

no comment

Reviewer 3 ·

Basic reporting

Accept

Experimental design

Authors attended to the suggestions of the reviewers.

Validity of the findings

Authors attended to the suggestions of the reviewers.

Additional comments

Authors attended to the suggestions of the reviewers.

·

Basic reporting

The author has written in simple, unambiguous, and competent Language. The literature is well-referenced and applicable, with an introduction and history to show meaning. The paper discusses a new method for LP recognition that involves stacking two CNN networks. Batch normalization and a non-Linear activation layer follow the convolution layer, which is at the core of the convolution block.

Experimental design

All of the research questions have been presented and answered in a clear and concise manner, and they are all important and significant in the context.

Validity of the findings

All validation problems have been addressed; they are stable, statistically accurate, and well-controlled.

Additional comments

I congratulate the authors on their extensive data collection. Furthermore, the manuscript is written in plain, unambiguous language, and the comments have been satisfactorily addressed.

---

## Author Rebuttal · Round 0.3

Electrical-Electronics Engineering

Bartin University – Turkey                                      March 4, 2021

Dear Editors,

We thank the editors and reviewers for their efforts, time and useful comments.

We have edited the manuscript addressing your valuable comments. We believe that your comments made the manuscript scientifically suitable to be published.

Mohammed Salemdeeb

PhD, Electronics and Communication Engineering.
Bartin University, Turkey,
Faculty of Engineering,
Electrical-Electronics Engineering department.

On behalf of all authors.

**Reviewer 3** (Anonymous)

Basic reporting
A deeper error analysis is required.

More error analysis sentences are added at line 323, 324, 330 and 347.

Update at line 323,
"All of the previous datasets were divided into training and test sets by their authors where the instances in the test set were collected from a different source (different writers for CR and different photographers for fashion) from the training set's source. The performance evaluation is done based on CNN type (stacked is simpler than spars), number of layers, number of learnable parameters and recognition error."

Other updates are written below.

Experimental design
The authors need to discuss the evaluation procedure.

All the reported results were concentrated on the classification error, the datasets were divided into training and test sets and the evaluation is based on CNN type, classification error, number of layers and number of learning parameters.
More sentences to discuss the evaluation procedure are added at lines 324, 330 and 347.

Validity of the findings
The authors can discuss the results for other performance metrics.

We discuss the results for performance metrics classification error, number of learnable parameters, number of layers and memory usage of FDCNN. In the previous related work these are the most common used metrics and we follow them to compare our performance and introduced dataset. Some of datasets were not entered to the training process but it was used for test, Zemris dataset was not seen in the training and the model was trained on other datasets but the model was tested only on Zemris, which reflects the validity of the results. More sentences are added to discuss the validity of the results at line 324, 330 and 347.

Comments for the Author
The literature review needs to be updated with some of the recent works. Discussing the following works would make the manuscript richer for the readers:

DevNet: An Efficient CNN Architecture for Handwritten Devanagari Character Recognition. Int. J. Pattern Recognit. Artif. Intell. 34(12): 2052009:1-2052009:20 (2020)
Character recognition based on non-linear multi-projection profiles measure. Frontiers Comput. Sci. 9(5): 678-690 (2015)

Relative Positioning of Stroke-Based Clustering: a New Approach to Online Handwritten Devanagari Character Recognition. Int. J. Image Graph. 12(2) (2012)
Artistic Multi-character Script Identification Using Iterative Isotropic Dilation Algorithm. RTIP2R (3) 2018: 49-62
Character Recognition Based on DTW-Radon. ICDAR 2011: 264-268
Spatial Similarity Based Stroke Number and Order Free Clustering. ICFHR 2010: 652-657
Dtw-Radon-Based Shape Descriptor for Pattern Recognition. Int. J. Pattern Recognit. Artif. Intell. 27(3) (2013)

Thanks for these comments, we believe that:
"DevNet: An Efficient CNN Architecture for Handwritten Devanagari Character Recognition. Int. J. Pattern Recognit. Artif. Intell. 34(12): 2052009:1-2052009:20 (2020)" is recent and related to our research regarding both character recognition problem and CNN. We have updated the literature review citing recent "DevNet" reference.

## Reviewer 4 (Kanika Thakur)

Basic reporting
A new approach to LP identification by stacking two CNN networks is discussed in the paper. The core of the convolution block, which is the convolution Layer is followed by batch normalization and a non-Linear activation layer.

Experimental design
There is a well-designed experimental section. The manuscript's strongest achievements are the suggested Full Depth CNN (FDCNN) model and the recommendation for a new license plate data set called LPALIC. The strength of the paper is the selection of parameters, filters, and the process of training. The manuscript also gives an overview of FDCNN with respect to the use of memory.
The drawback of the manuscript is that the results obtained during the testing process on the test dataset are missing and no validation dataset has been used, so it is unclear that how fine-tuning of the model hyperparameters has been performed and the problem of overfitting has been addressed. I think the writers should be able to update the manuscript before publishing to make all these points clear and further illustrate and solidify their already good findings

Thank you, we did a lot of experiments and the written results are the final average results. Since we have 11 tables in our paper and we see that we cannot add more tables otherwise the paper will be too long.
Most of the tests done on LPALIC and ALPR datasets were designed to overcome the overfitting problem by training the model on a dataset and testing the model on another characters dataset.

For example, to test UCSD dataset we trained the model on our Turkish and EU LP characters set. FDCNN is tested on many datasets and the average performance is very good on test sets. Fine-tuning of the model hyperparameters has been performed as described at line 284 by retraining the model again using other training algorithm and different training settings. When we did, the results get better which may be seen as a low effect of the overfitting problem.

We updated our manuscript to make all these points clear and further illustrate and solidify our findings at lines 324, 330 and 347.

Update at Line 324 "However, Khaled et al., (2010) used his dataset for both training and testing, FDCNN could classify the whole dataset (as a test set) of (Khaled et al., 2010) with error of 0.46\% whereas the training was done on characters collected and cropped manually from public KSA LP images."

Update at line 330 "Zemris, UCSD, Snapshots and ReId datasets were not used in the training process but the proposed FDCNN was tested on each of them as a test set to ensure that the model was fitted to character features, not to a dataset itself. For UFPR dataset, FDCNN was tested two times on UFPR test set, training on only the training set of UFPR and training on both UFPR and LPALIC characters."

Update at line 347 "In the manual split in Table 11, the country's characters training and testing sets were used to train and test FDCNN. In trained on other countries, the FDCNN was trained on both the country's characters training set and other countries characters but tested only on that country's test set. In the random 80/20 split, the country's characters were split randomly into training and testing sets, and FDCNN was trained on both the split country's characters training set and other countries characters but tested only on that split country's test set, a lot of random split tests were done and the average errors were reported in the table. Those different test analyses were done to validate and evaluate the results and reduce the overfitting problem."

Validity of the findings
All outcomes are well defined and codes are given, but to ensure the validity of the data, the points listed must be answered. I assume that the relevance of the results can be more readily asserted if the issues are discussed and explained.

Comments for the Author
The paper is clear, but there are few points that require Clarification
1. Explains the methodology used in order to correctly recognize Arabic zero numbers and letters written in continuous style.

Thank you for this comment. This paper is interested in isolated character recognition. However, the Arabic words are written in a continuous style but in this study we used the isolated Arabic character datasets.

At line 66 we described the solution proposed by Abdleazeem and El-Sherif (2008), in our paper we used the same logic of size-sensitive feature by zero character half size reduction since it has a smaller size than other characters.

A new explanation sentence is added at line 303," The same logic of size-sensitive feature proposed in (Abdleazeem and El-Sherif, 2008) is used to solve the problem of Arabic zero character by half size reduction for Arabic zero character image (in MADbase dataset) since it has a smaller size than other characters."

2.Discuss the criteria based on which a character is labeled as difficult /easy as specified in line number 342.

Agreed, the criteria here is difficulty of manually labelling the character images in the dataset preparing stage (labelling by human).

A clarification parentheses is added at line 342. (difficult at manual labelling the character images in the dataset preparing stage)

3.You should clarify, why one should go for the proposed FDCNN model, when Assiri,2019 stacked method has outperformed.

At line 297, we wrote a simple clarification.
We edited the clarification paragraph at line 297 to be like this:
"However, the architecture in (Assiri, 2019) has 15 layers with 13.12M parameters while FDCNN has 12 layers with 1.69M parameters which means that FDCNN is simpler and 7 times faster (in terms of the number of parameters 13.12/1.69). The results in Assiri 2019 were obtained utilizing data augmentations (not used in FDCNN training), different training processes (FDCNN training process is simpler as described in the previous section) and Dropout layers before and after each pooling layer with different settings, but, FDCNN has no Dropout layer and showed good results on MNIST."

---

## Round 0.4 · accepted · Accept

Well revised, and I am happy to consider this paper for publication. Congratulations!